# DynaPipe: Dynamic Layer Redistribution for Efficient Serving of LLMs with Pipeline Parallelism

**Hongxin Xu**[*1]**, Tianyu Guo**[*1]**, Xianwei Zhang**[†1]
[1]CSE, Sun Yat-Sen University
{xuhx56,guoty9}@mail2.sysu.edu.cn,zhangxw79@mail.sysu.edu.cn

## Abstract

To accelerate large language model (LLM) inference, pipeline parallelism partitions model layers into sequential stages, each assigned to a different device for concurrent execution. However, this method often suffers from pipeline bubbles caused by imbalanced computation in the tail stage. While upstream stages focus solely on layer-forward operations, the final stage must also handle additional post-processing tasks like sampling, which introduces significant latency. This discrepancy in workload leads to pipeline misalignment, forcing upstream stages to idle and degrading overall performance. Existing frameworks typically distribute layers evenly across stages without accounting for computational load differences. To address this, we propose `DynaPipe`, a dynamic layer redistribution scheme that adaptively balances computation by predicting execution latency in real time. Moreover, we introduce an asynchronous key-value (KV) cache migration coordinator to enable non-blocking layer redistribution during inference. Experiments on representative LLMs demonstrate that `DynaPipe` reduces average end-to-end request latency by 8% to 41% across diverse workloads, outperforming state-of-the-art pipeline parallelism systems. Our implementation is publicly available at `https://github.com/xhx1022/DynaPipe`.

## 1  Introduction

The rise of large language models (LLMs) [1, 2, 3] has become a cornerstone of modern artificial intelligence, empowering a wide range of application fields, from conversational agents [4, 5] and intelligent coding assistants [6, 7] to retrieval-augmented generation systems [8, 9, 10]. Despite their impressive capabilities, deploying LLM at scale presents significant challenges, primarily due to their immense computational and memory demands. As model sizes grow from billions to hundreds of billions of parameters, inference costs rise exponentially, rendering single-device execution impractical. This has spurred extensive researches and engineering efforts aimed at developing efficient distributed inference strategies [11, 12, 13, 14, 15].

Among these strategies, Tensor Parallelism (TP) [12, 16, 17] and Pipeline Parallelism (PP) [18, 19, 20] have emerged as two dominant methods, now widely adopted in industrial-scale LLM serving systems [21, 22]. TP partitions the computation within each layer across multiple devices, but its scalability can be often restricted by substantial inter-device communication overhead, particularly in bandwidth-constrained scenarios. In contrast, PP divides the model layers into sequential stages, each assigned to a different device. Thus, PP processes multiple micro-batches in a pipelined manner, improving throughput by reducing communication overhead and enabling better overlap of computations across stages.

---

[*]Equal contribution
[†]Corresponding author.

39th Conference on Neural Information Processing Systems (NeurIPS 2025).

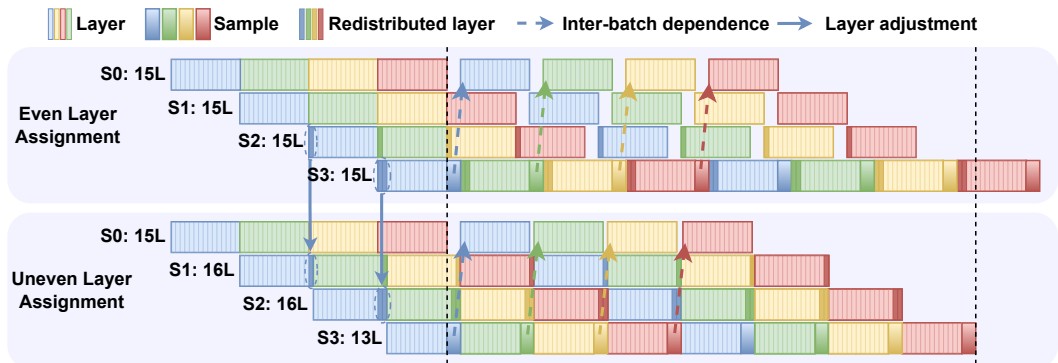

Figure 1: Illustration of pipeline bubbles during autoregressive decoding. Top: Conventional even layer allocation leads to idle upstream stages due to tail-stage sampling delays. Bottom: Uneven layer assignments to reduce idle time and improve hardware efficiency. "S0:15L" means stage 0 consists of 15 layers.

While PP has proven effective in low-bandwidth environment, it introduces a critical computational overhead that has not been adequately addressed: the additional latency introduced by the sampling[3] operation at the final stage of the pipeline. As shown in Figure 1, traditional pipeline systems typically distribute model layers evenly across stages but fail to account for the workload imbalance caused by sampling computation. In autoregressive generation, each decoding step strictly depends on the previous one, meaning the next step can only begin after the current token is generated. This dependency forces upstream stages to wait for the output from the final stage, resulting in significant pipeline bubbles that reduce hardware utilization. As sampling becomes more time-consuming, it bottlenecks the entire pipeline's progress. The inherent stage dependency exacerbates this issue, as each stage must wait for the previous one to complete. Consequently, the sampling latency propagates through all stages, amplifying the bubble effect and degrading overall execution efficiency.

To address these challenges, we propose DynaPipe, a solution that dynamically reallocates model layers across pipeline stages during runtime to optimize workload balance. As illustrated in the bottom of Figure 1, DynaPipe redistributes some layers from the final stage to earlier pipeline stages, thereby reducing computational imbalance and significantly minimizing pipeline bubbles. To achieve this, DynaPipe continuously monitors the runtime status of each stage and dynamically adjusts the layer-to-stage mapping to balance the forward computation and sampling load. Moreover, DynaPipe avoids pipeline stalls during redistribution by introducing an asynchronous KV cache migration mechanism, which enables seamless and non-blocking layer redistribution across devices.

In summary, the contributions of this paper are as follows:

- We identify a critical pipeline efficiency bottleneck in LLM inference that has been largely overlooked. The sampling operation at the last stage creates severe pipeline bubbles and significantly reduces GPU utilization.

- We propose DynaPipe, a dynamic layer redistribution scheme that adaptively optimizes layer allocation across pipeline stages at runtime to balance forward computation and sampling workloads. By introducing an asynchronous KV cache migration mechanism, DynaPipe enables non-blocking pipeline adjustments for seamless execution.

- Our evaluation of DynaPipe across diverse data conditions demonstrates its consistent superiority over state-of-the-art frameworks, reducing average end-to-end latency by 8–41% while enhancing service-level objective (SLO) attainment.

---

[3]For convenience, we collectively refer to the calculation of logits and the sampling operation as "sample".

## 2 Background and Motivation

### 2.1 Transformer-based LLM Inference

LLMs based on the Transformer architecture [23] perform autoregressive generation by stacking multiple decoder layers. Each decoder layer primarily comprises a self-attention mechanism and a feed-forward network (FFN). During inference, each input token is first converted into a continuous vector through an embedding layer and then passed sequentially through the decoder layers. Within each layer, the model computes the query (Q), key (K), and value (V) vectors. The attention mechanism calculates the similarity between Q and K via dot product, applies a softmax function to obtain normalized weights, and uses them to compute a weighted sum over V, thereby capturing dependencies between tokens at different positions. The FFN then applies fully connected layers with non-linear activation to enhance the model's representational capacity. The final hidden state is then projected to logits, representing a probability distribution over the vocabulary. The next token is sampled based on this distribution using strategies such as greedy decoding [24] or beam search [25].

To improve inference efficiency, modern LLM incorporates a KV cache mechanism. During autoregressive generation, a new Q attends to past K and V at each step. Since K and V remain unchanged, caching them avoids redundant KV computation and improves inference efficiency. Let $n$ denote the number of tokens to be computed at the current step, $L$ the total sequence length, and $d$ the dimension of the hidden states. The total computational complexity of the attention mechanism is $O(nd^2 + nLd)$, which includes the cost of computing Q, K, V and the attention operations. The complexity of the FFN is $O(nd^2)$. By reusing cached K and V, the KV cache significantly reduces the time and memory overhead of attention, especially in long-sequence generation tasks, thereby improving inference speed and overall efficiency.

### 2.2 Parallelism in LLM Inference

To achieve efficient distributed inference for LLM, modern systems typically employ two mainstream parallelization strategies: TP and PP. TP divides the large matrix operations within a model layer and distributes them across multiple devices for parallel computation, overcoming the memory limitations of a single GPU. However, this fine-grained parallel strategy requires frequent exchange of intermediate activations during the model's forward pass, leading to significant cross-device communication overhead.

Alternatively, PP employs a coarse-grained model partitioning strategy, which divides the model into sequential stages of consecutive layers, with each stage being assigned to different devices for execution. During inference, input data is split into multiple micro-batches, which are fed into the pipeline in a time-sequential manner and processed through each stage. Once completing computation on a micro-batch, a pipeline stage asynchronously transmits the intermediate outputs to the next stage, and immediately proceeds to process the next micro-batch. As micro-batches flow concurrently across different stages, the system forms an efficient pipeline scheduling mechanism, significantly improving computational resource utilization and overall throughput. Since PP only requires data exchange between adjacent stages, it substantially reduce communication bandwidth pressure. As a result, PP has become a core parallelization technique widely adopted in current industrial-grade LLM inference systems.

### 2.3 Challenges in PP

Inspired by TD-Pipe [26] and gLLM [18], we find that although PP provides notable efficiency benefits for LLM inference, its performance remains susceptible to inter-stage bubbles. These bubbles typically arise from imbalanced computational workloads across different pipeline stages, leading to inefficiencies and idle time. Specifically, while some stages are still processing time-consuming tasks, others remain idle, waiting for results. Through detailed performance analysis, we have identified a critical but previously overlooked bottleneck caused by the sampling latency at the end of each model forward.

While most pipeline stages perform uniform forward computations, the final stage necessitates to handle additional token sampling operation, which adds extra computational load. This imbalance makes the final stage a potential performance bottleneck, as the additional sampling workload disrupts the even distribution of computational tasks across stages. Although static layer redistribution

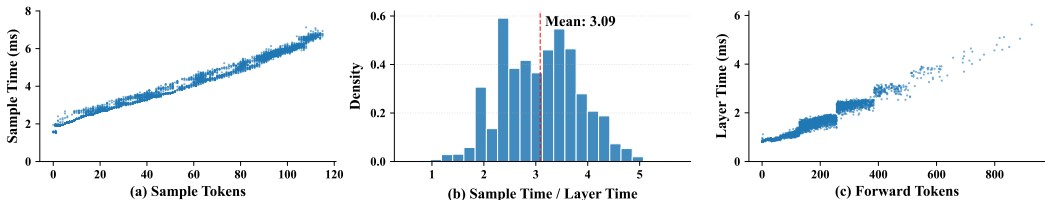

Figure 2: The overhead analysis of sampling and forwarding.

can partially mitigate the sampling overhead, the dynamic nature of both sampling and forward computation times makes a single static strategy unable to maintain balanced workloads under dynamic workload conditions. The fundamental reason is that sampling overhead mainly depends on the number of tokens to be sampled, which is determined by the number of decode requests, since prefill tokens do not require sampling.

Figure 2 presents profiling results that analyze this overhead in detail. As shown in Figure 2(a), sampling latency grows approximately linearly with the number of sampled tokens, indicating that increased decode requests directly lead to higher sampling overhead. Meanwhile, Figure 2(c) illustrates the relationship between the forward computation time of a single layer[4] and the total number of tokens, showing that forward computation time is also dynamic and positively correlated with the combined number of prefill and decode tokens in the batch. Consequently, the ratio of sampling time to single-layer forward time is not constant, but varies with the composition of tokens in each batch, reflecting the dynamic proportion of decode to prefill tokens. As shown in Figure 2(b), this ratio approximately follows a normal distribution. Statistical results further indicate that, on average, the sampling overhead is about 3.09× the latency of a single layer's forward computation. This finding highlights that the sampling operation is not a negligible terminal step but rather a non-trivial performance factor that introduces substantial latency and significantly impacts the overall efficiency of the pipeline.

In summary, current PP-based inference systems lack the ability to dynamically adapt to the fluctuating workload introduced by sampling. Consequently, sampling-induced bottlenecks at the tail stage can propagate upstream, negatively impacting overall system performance. To address this issue, a dynamic workload balancing mechanism is needed that reallocates computational workloads to reduce sampling-induced bubbles and improve overall efficiency of the pipeline.

## 3 Design

`DynaPipe` is a collaborative optimization framework designed to reduce pipeline bubbles caused by tail-stage sampling. As illustrated in Figure 3, `DynaPipe` consists of three core components: an execution time predictor, a bubble-aware scheduler, and a migration coordinator. The predictor employs a lightweight predictive model to estimate the per-layer forward computation time and the sampling latency. Using these estimates, the scheduler determines whether dynamic layer redistribution is needed to balance the workload across pipeline stages. When migration is triggered, the system asynchronously transfers KV cache across devices ensuring that the pipeline execution continues without interruption. By tightly coordinating these components, `DynaPipe` dynamically adapts to runtime workload variation, effectively minimizing pipeline inefficiencies and improving hardware utilization.

### 3.1 Execution Time Predictor

As described in Section 2.3, the latency of forward and sampling computations in LLM inference varies significantly at runtime. To develop an effective layer redistribution strategy, accurate estimation of latency at each stage is essential. To this end, we construct two independent latency prediction models. The first model estimates the forward computation time of a single Transformer layer based on the number of tokens $n$ processed per step and the sequence length $L$. The second model estimates

---

[4]With identical architecture and batch, per-layer forward time is uniform and serves as a standard measure for sampling cost, guiding the subsequent determination of the number of layers to reallocate.

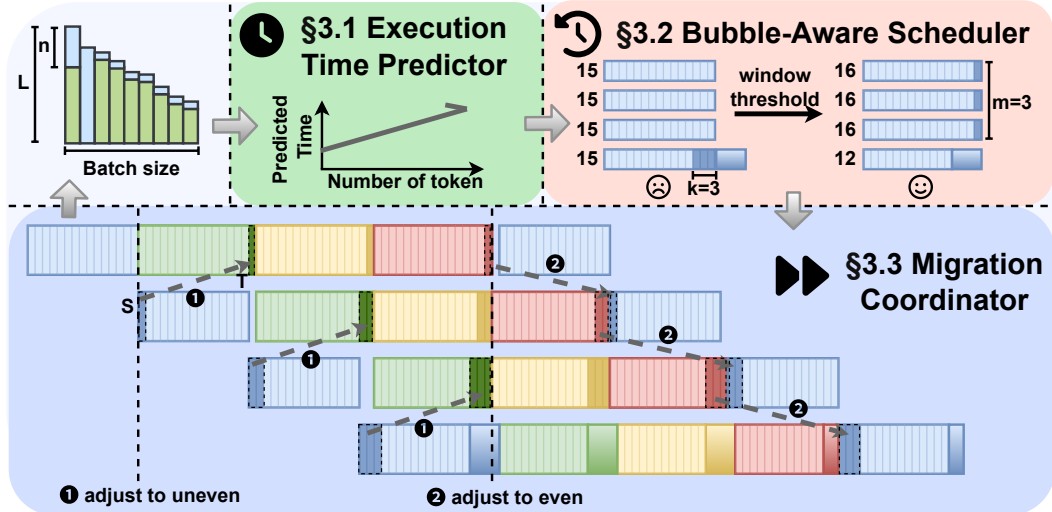

Figure 3: Overall architecture and workflow of `DynaPipe`. "S/T" means source/target stage.

the sampling overhead, which increases approximately linearly with the number of sampled tokens, as shown in Figure 2(a). Together, these models form our execution time predictor, capable of dynamically estimating per-layer inference latency and sampling cost, enabling an accurate and adaptive pipeline layer redistribution strategy.

To construct a latency prediction model with theoretical grounding, the analysis starts with the computational characteristics of a single Transformer layer. As discussed in Section 2.1, the computation of a single layer primarily consists of the QKV projection, self-attention mechanism, and the feed-forward network (FFN). Since the computational complexity of the QKV projection and FFN is the same, they can be combined, simplifying the overall computational complexity of the layer as $O(nd^2 + nLd)$.

For practical deployment, we adopt a scheduling strategy known as chunked prefill[19], which merges prefill and decode requests within the same computation chunk. The prefill phase is compute-intensive, as it processes the full input sequence in a chunk. In contrast, the decoding step generates one token at a time, relies heavily on KV cache access, and is thus constrained by memory bandwidth. Despite their different computational characteristics, both phases are executed together in a unified forward computation. This unified computational flow motivates a joint modeling approach to capture the latency characteristics of both phases. Notably, in the decode phase, $n$ represents the number of tokens to be computed per forward pass, which is typically 1.

We model the total execution latency of a single Transformer layer over all requests within a chunk as:

$$T_{\text{layer}} = \sum_{i=1}^{N} (\phi_1 n_i + \phi_2 n_i L_i + \epsilon) \tag{1}$$

Here, $\phi_1$, $\phi_2$ and $\epsilon$ are parameters fitted from profiling data. $N$ denotes the number of requests contained in the chunk. This formulation accounts for request-level computational costs, providing a practical estimate of total layer latency under chunked execution.

In addition, to account for the sampling overhead in the tail stage of decoding, we introduce a separate prediction model:

$$T_{\text{sample}} = \alpha \cdot N_{\text{decode}} + \beta \tag{2}$$

where $N_{\text{decode}}$ denotes the number of decode requests in the chunk, and $\alpha$, $\beta$ are fitted coefficients representing the growth rate and fixed overhead of sampling, respectively.

## 3.2 Bubble-Aware Scheduler

Building on the latency estimates provided by the execution time predictor, pipeline stages can be assessed for potential imbalances during LLM inference. To address these imbalances, we design a bubble-aware scheduler that dynamically adjusts layer allocation across stages. The scheduler aims to minimize pipeline bubbles resulting from uneven execution times, thereby improving overall pipeline efficiency and hardware utilization.

In particular, as the number of decode requests increases and more tokens need to be sampled, the sampling overhead in the final pipeline stage becomes non-negligible. To ensure that this stage maintains the same computation time as other stages and further to achieve pipeline alignment, we reduce the number of layers assigned to this final stage. Let $k$ denotes the number of layers removed from this stage and $m$ denotes the number of stages received these layers. These removed layers are evenly redistributed to the upstream pipeline stages to maintain workload balance.

To quantify the misalignment introduced by this adjustment, we define the following metric:

$$\Delta = \left| T_{\text{sample}} - k \cdot T_{\text{layer}} - \frac{k}{m} \cdot T_{\text{layer}} \right|, \quad m = \min(k, \text{num}_{\text{stages}} - 1) \tag{3}$$

This metric measures the difference in computation time between the final stage and the preceding stages after layer redistribution. Ideally, when $\Delta \approx 0$ and $m = num_{stages} - 1$, all pipeline stages are aligned in execution time, leading to improved pipeline efficiency and resource utilization.

To further enhance the stability of the scheduling strategy and avoid the overhead caused by frequent redistribution, we introduce a stability window threshold mechanism. A configuration is activated only when it appears consistently across all positions within a sliding window, effectively filtering out short-term fluctuations. This mechanism enhances scheduling robustness while ensuring smooth and efficient scheduling during inference.

## 3.3 Migration Coordinator

During the initialization phase, the system pre-allocates memory and loads the weights for additional layers that may be reassigned. This enables pipeline stages to quickly access these preloaded parameters during redistribution. However, weight preloading alone does not complete the migration process, as it also requires the transfer of the corresponding KV cache. To address this, we designed and implemented a migration coordinator to manage and transfer KV cache during layer redistribution, thereby enabling efficient collaboration across different pipeline stages.

As shown in Figure 3, when the scheduler triggers layer redistribution, the migration coordinator minimizes disruption to ongoing inference, enabling a smooth transition and preserving overall system performance. The core idea behind this design is to leverage pipeline parallelism and maximize the overlap between computation and communication during the migration process. Specifically, when the layer configuration of a stage changes, the system compares the current and adjusted layer assignments to identify which layers need to be migrated or received, thereby determining the precise migration scope. During migration, once the source stage completes computation for the layers to be migrated, it asynchronously transmits the corresponding KV cache to the target stage. Meanwhile, the target stage asynchronously receives the KV cache for the layers it is about to take over and continues executing forward computations for the unaffected layers. The target stage only waits for the KV cache when it reaches the newly assigned layers, and begins computation as soon as it receives the cache for those layers. Since these KV cache are typically sent in advance by the source stage, the target stage can complete reception promptly, avoiding prolonged stalls. This design achieves parallel overlap of computation and communication, avoiding prolonged pipeline stalls caused by migration operations and ensuring a quick switch in layer redistribution.

# 4 Evaluation

## 4.1 Experimental Setup

**Implementation.** We implement `DynaPipe` based on gLLM [18], a compact and efficient LLM inference framework that outperforms existing production-level inference frameworks in pipeline parallelism. The pipeline workers use ZeroMQ [27] for efficient inter-process communication,

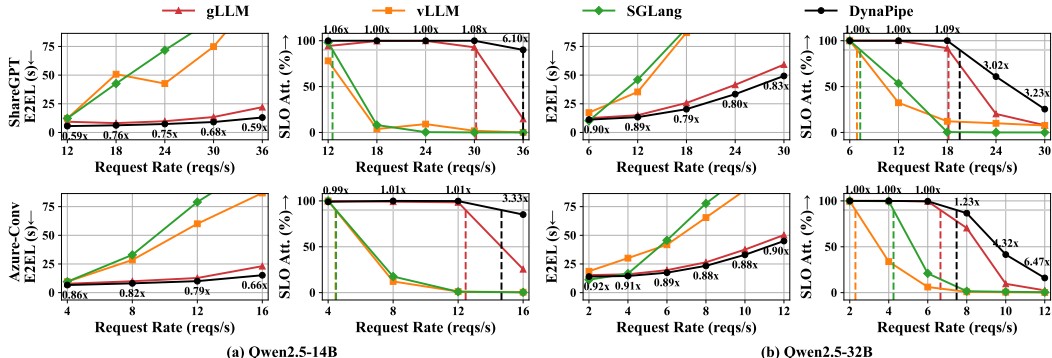

Figure 4: The average latency and SLO attainment rate of different LLM serving systems. `DynaPipe` achieves the lowest average latency and the highest SLO compliance across all workloads.

ensuring smooth synchronization between stages, with each worker responsible for specific tasks. KV cache migration is performed using NCCL [28] for high-performance cross-GPU communication, optimizing synchronization and data transfer in multi-GPU environments. All experiments are conducted on a system with four NVIDIA A100-PCIe-40GB [29] GPUs connected via PCIe. For `DynaPipe`, we set the window threshold size to 25 to strike a balance between performance and overhead.

**Datasets and models.** We use the ShareGPT [30] and Azure-Conv [31] datasets to evaluate the system's performance. The ShareGPT dataset, which contains real-world conversational data between users and ChatGPT [3], is widely used for evaluating dialogue systems. In contrast, the Azure-Conv dataset, sourced from Azure's production environment for dialogue inference services, typically shows longer input lengths than output lengths. Table 1 summarizes the average input and output lengths for both datasets. We perform experiments using two different model variants: Qwen2.5-14B and Qwen2.5-32B [1].



Table 1: Input / Output length.

| Dataset | | #Input | #Output |
|---|---|---|---|
| ShareGPT | P50 | 221 | 157 |
| | P90 | 627 | 382 |
| Azure-Conv | P50 | 514 | 192 |
| | P90 | 1008 | 412 |

Table 2: SLO requirements.

| Dataset | Model | TTFT (s) | TPOT (ms) |
|---|---|---|---|
| ShareGPT | 14B | 1 | 100 |
| | 32B | 4 | 250 |
| Azure-Conv | 14B | 1 | 100 |
| | 32B | 4 | 150 |



**Baselines.** We compare `DynaPipe` with several state-of-the-art LLM inference frameworks, including vLLM (v0.8.5 V1) [21] and gLLM [18], both using pipeline operations, as well as SGLang (v0.4.3.post2) [22], which employs tensor parallelism. All these systems use the chunk prefill technique [19], with the maximum chunk size set to 2048, serving as the baseline for performance evaluation.

**Metrics.** We evaluate system performance using two metrics: the average end-to-end latency per request (E2EL) and the SLO compliance rate. E2EL refers to the total time from when a request is issued to when the response generation is completed. The SLO compliance rate measures the system's stability under service-level constraints: a request is considered compliant if both its TTFT and TPOT are below the thresholds listed in Table 2. TTFT is the time from request submission to the generation of the first token; TPOT is the average time to generate each output token. The SLO compliance rate is defined as the proportion of requests that meet both latency constraints.

## 4.2 Overall Performance

We conduct a comprehensive evaluation of `DynaPipe` across multiple datasets and model scales. Experimental results demonstrate that `DynaPipe` consistently outperforms mainstream inference frameworks under various workloads. As illustrated in Figure 4, the performance gap is particularly

notable when compared with vLLM and SGLang, both of which fall significantly behind gLLM due to architectural limitations. Specifically, SGLang relies solely on tensor parallelism, which incurs substantial communication overhead under high-concurrency conditions, quickly becoming a performance bottleneck. vLLM, while adopting pipeline parallelism to improve throughput, uses fixed-size chunks that lead to pronounced inter-batch pipeline bubbles, resulting in suboptimal hardware utilization. gLLM mitigates these inter-batch bubbles via more adaptive scheduling but overlooks the non-trivial overhead introduced by sampling, leading to inter-stage bubbles that further constrain end-to-end performance.

In contrast, `DynaPipe` redistributes layers to enable sampling to overlap with the forward computation of extra layers in other stages. This design effectively fills idle phase in the pipeline and alleviates structural pipeline bubbles, yielding a substantial performance boost. As shown in Figure 4, `DynaPipe` consistently achieves superior performance under varying request loads, with especially pronounced gains under high-throughput scenarios. This improvement stems from earlier completion of the prefill phase at higher loads, resulting in an increased proportion of decode requests per batch. Consequently, the sampling overhead becomes a more significant bottleneck, which `DynaPipe` is well-positioned to mitigate.

Furthermore, experiments across different model sizes show that `DynaPipe` consistently reduces end-to-end latency, highlighting its effectiveness in mitigating sampling overhead. On the ShareGPT dataset, `DynaPipe` achieves up to a 40% reduction in E2EL, while on Azure-Conv, the reduction reaches 34%. The relatively smaller gain on Azure-Conv is attributed to its higher input-output length ratio, which leads to longer per-layer execution time in the prefill phase and reduces the relative impact of sampling. Nevertheless, `DynaPipe` still delivers substantial improvements in this scenario, highlighting its versatility and resilience under varying workload conditions and data distributions.

In the SLO attainment analysis, `DynaPipe` demonstrates significant advantages. Under the 90% SLO attainment target, SGLang and vLLM struggle to meet the requirements at high request rates. In contrast, `DynaPipe` maintains higher SLO compliance under high request rates. `DynaPipe` can also sustain up to 19% higher request rates than gLLM under 90% SLO attatinment.

## 4.3 Performance under Different Output-Input Length Ratios

The acceleration effect of `DynaPipe` is closely related to the output-input length (O:I) ratio. We thus test various static redistribution strategies on a synthetic dataset, with the input length fixed at 512 and the output length varying to form different O:I ratios. As shown in Figure 5, during the prefill phase (output length = 1), sample overhead is minimal. In this case, static average distribution achieves the lowest latency, with `DynaPipe` performing similarly. In contrast, other static redistribution strategies are unnecessary at this case and instead introduce inappropriate layer assignments, leading to inter-stage load imbalance and degraded performance. As the output length increases, the decoding phase becomes significantly longer, leading to a gradual rise in sampling overhead. In this scenario, static load-balancing strategies fail to address the resulting inter-stage imbalance, ultimately causing a decline in hardware utilization. On the other hand, the performance of other static layer redistribution strategies begins to improve, mainly because adding additional layers helps offset the sample overhead, balancing the pipeline. However, the effectiveness of different static layer redistribution strategies varies under different input and output length ratios. The greater the proportion of sample overhead, the more layers are allocated, which is more beneficial. Nevertheless, `DynaPipe` consistently outperforms other methods with the lowest latency, and its adaptive capabilities make it especially effective in real-world scenarios where input and output lengths fluctuate.

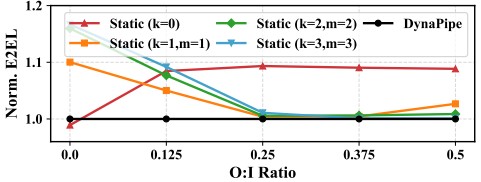

Figure 5: Normalized E2EL under different O:I ratios with various adjustment strategies.

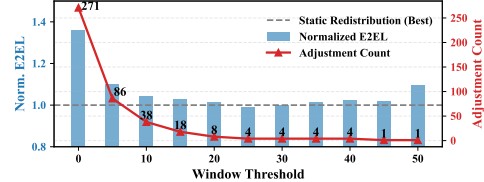

Figure 6: The effect of adjustment window threshold on E2EL and adjustment frequency.

## 4.4 Performance under Different Window Thresholds

Figure 6 normalizes the E2EL using the best static layer redistribution strategy on the ShareGPT dataset as the baseline, illustrating the impact of the adjustment window threshold on the E2EL and adjustment count for the 32B model. It can be observed that when the adjustment window threshold is too small, the scheduler frequently triggers re-adjustments, leading to significant overhead, with the end-to-end latency increasing by 35% compared to the best static redistribution strategy. On the other hand, if the window threshold is too large, the scheduler fails to adjust in time to address the overhead caused by sample operations, resulting in an increase in end-to-end latency. Setting the window size to 25 strikes a good balance between these factors. As shown in the figure, on the ShareGPT dataset, only 4 adjustments were made, but the end-to-end latency is comparable to the best static layer redistribution strategy.

## 4.5 Performance under Multi-Node Setting

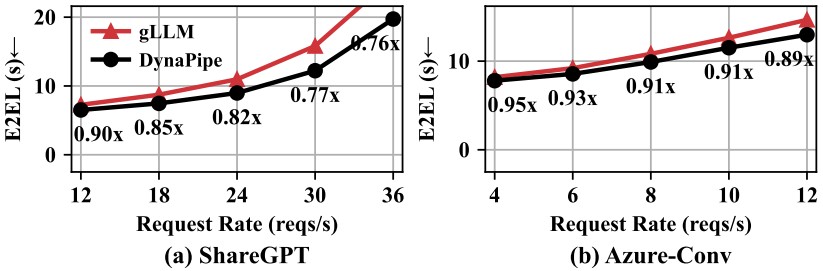

Figure 7: The latency comparison between `DynaPipe` and gLLM on Qwen2.5-14B (4 nodes).

To emulate a realistic cross-node deployment scenario, we configure NCCL by setting NCCL_SHM_DISABLE=1, NCCL_P2P_DISABLE=1, and NCCL_IB_DISABLE=1, thereby disabling shared memory, peer-to-peer, and InfiniBand communication backends. This setup forces all inter-GPU communication to go through the TCP stack, effectively simulating network latency and bandwidth characteristics of distributed environments. All experiments are conducted on the Qwen-14B model with four NVIDIA A100 GPUs. As shown in Figure 7, `DynaPipe` achieves significant performance improvements compared to gLLM. Although cross-node communication introduces additional KV Cache migration overhead, which slightly narrows the optimization margin compared with the single-node scenario, the overall performance remains at a high level. This demonstrates that the core mechanism of `DynaPipe` maintains strong effectiveness and scalability in cross-node settings.

## 4.6 Effect of Execution Time Predictor

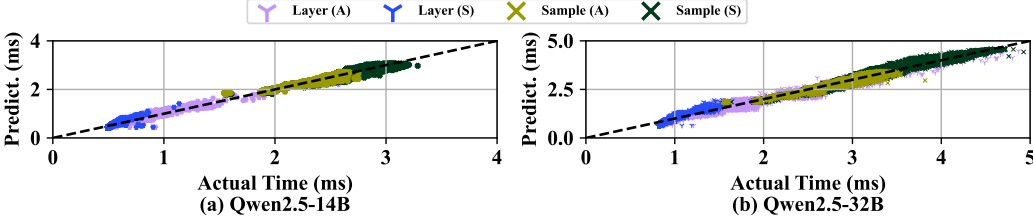

Figure 8: Actual vs. predicted execution time distribution. "A/S" means Azure-Conv/ShareGPT.

We construct a synthetic dataset by randomly sampling input and output sequence lengths and measured the offline execution times of the 14B and 32B models to fit the corresponding execution time predictors. In Figure 8, we evaluate the accuracy of the predictors for both layer execution time (Layer) and sampling execution time (Sample). The results show that the predicted values closely align with the y=x line, indicating a high consistency between the predicted and actual execution times. The average relative error for layer execution time is 4.95%, while the error for sampling execution time is only 0.31%. This demonstrates that the predictors fitted for the same LLM maintain

high accuracy across different dataset scales, validating the generalization ability of our model. Furthermore, the predictor takes only 0.5 microseconds per prediction, with negligible overhead.

## 5   Related Work

**LLM inference optimization.**   Recent system-level research has increasingly concentrated on enhancing the memory efficiency and throughput performance of LLM inference. Representative efforts include vLLM [21], which alleviates GPU memory fragmentation through the PagedAttention mechanism, and SGLang [22], which improves execution efficiency by exploiting structured execution and KV cache reuse. In addition, several studies [32, 33] employ performance modeling to guide scheduling strategies, thereby decoupling the prefill and decode phases to achieve higher GPU utilization. Within distributed inference settings, Llumnix [34] and Seesaw [35] further address load balancing through KV cache migration and adaptive re-sharding. Orthogonal to these works, DynaPipe focuses on a previously underemphasized issue of stage imbalance in pipeline parallelism, which adaptively mitigate stage imbalance at runtime and significantly improve overall inference efficiency.

**Pipeline Parallelism.** Extensive pipeline parallelism research in LLM training has primarily focused on communication latency optimization [36], memory balancing [37, 38, 39], pipeline bubble mitigation [38, 40, 41, 42], and activation checkpointing [37, 43]. Some studies have also identified inter-stage imbalance issues, with approaches such as Skywork-MoE [42] and Megatron-LM [44] employing static layer redistribution to alleviate this problem. However, such static strategies often prove less effective under the dynamic workload conditions that are typical in model serving. In addition, [41] proposes a Vocabulary Parallelism method designed for training scenarios to mitigate bubbles caused by sampling overhead during training. Nevertheless, applying this method directly to inference requires stricter synchronization, which significantly increases system design complexity and synchronization overhead, making it less suitable for efficient serving deployment. In LLM serving, several studies [19, 18, 26] have explored pipeline parallelism by tuning micro-batch size and balancing workload to reduce pipeline bubbles. However, these approaches have not fully addressed the inter-stage imbalance caused by uneven sampling overhead. DynaPipe complements these efforts by dynamically adjusting layer distribution based on real-time workload conditions, triggering redistribution only when significant changes occur. This design maintains relatively low communication overhead and significantly improves the overall efficiency of inference serving.

## 6   Limitation and Future Work

This work employs deliberate design trade-offs to improve overall system performance. Although the optimized runtime KV cache migration still incurs some communication overhead, it only results in minor local performance fluctuations, while overall system performance remains improved. In addition, to support potential future layer reallocations, extra GPU memory must be reserved for storing incoming weights, which reduces memory utilization efficiency. To address these issues, future work may incorporate compression techniques to reduce transfer overhead and leverage offloading mechanisms to minimize memory waste, thereby improving overall system performance. Furthermore, integrating vocabulary parallelism [41] into the inference pipeline is also a promising direction for future exploration.

## 7   Conclusion

In this paper, we investigate a significant but insufficiently explored aspect: the pipeline inter-stage bubble problem caused by sampling operations. We propose DynaPipe, a novel runtime dynamic layer redistribution scheme to address this challenge effectively. By dynamically adjusting the computational load across layers, DynaPipe achieves balanced task distribution among pipeline stages, effectively aligning the pipeline and mitigating inter-stage imbalance. Compared to state-of-the-art pipeline inference frameworks, DynaPipe delivers significant performance improvements.

## Acknowledgements

We sincerely thank the anonymous reviewers for their valuable comments and suggestions, which have greatly helped improve this work. This research is supported by the National Key R&D Program of China (Grant No. 2023YFB3002202), the NSFC grants (#62472462, #62461146204), and is sponsored by CCF-Tencent Rhino-Bird Open Research Fund (CCF-Tencent RAGR20240102).

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

# Appendix

## A    Performance Evaluation on Other Models

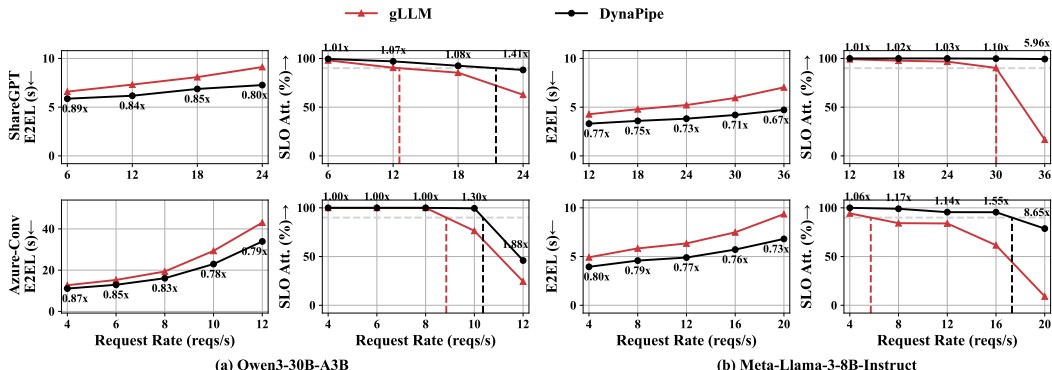

Figure 9: Performance comparison on MoE and Dense LLM models. DynaPipe consistently outperforms the baseline in both average latency and SLO compliance across all workloads.

As shown in Figure 9, DynaPipe consistently outperforms the baseline on both MoE and dense LLM models. The experiments are conducted on a single node equipped with four NVIDIA A100 40GB GPUs. The results reveal that, under pipeline parallelism, the inter-stage load imbalance caused by sampling overhead is a pervasive issue, regardless of the underlying model architecture. By dynamically detecting and adapting to such imbalance, DynaPipe effectively accelerates the inference process across different models. This further demonstrates the strong generality and scalability of the proposed method in a wide range of model scenarios.

## B    Analysis of System Overhead

**Migration Overhead Analysis** In our system, dynamic layer redistribution inevitably introduces a certain amount of KV cache migration overhead. Since the KV cache is transferred at the layer granularity, the communication cost of each migration remains relatively fixed under PCIe interconnect conditions. Meanwhile, the main latency of model forward computation is strongly correlated with the batch size: when the batch size is large, we overlap KV cache migration with part of the forward computation, effectively hiding most of the communication latency; when the batch size is small, although the overlap is less complete, the additional delay of a single layer migration can still be kept within 100 ms. Furthermore, when deployed in environments with high-bandwidth interconnects such as NVLINK, this overhead can be further reduced significantly.

To mitigate the disturbance caused by frequent migrations, we introduce a window-threshold mechanism into the scheduling policy. The system does not trigger redistribution upon every minor fluctuation in the ratio between sampling latency and single-layer forward computation cost. Instead, the signal is accumulated over a sliding window, and redistribution is only executed when the accumulated value exceeds a predefined threshold. This design effectively avoids oscillations near decision boundaries. For instance, when the sampling-to-forward ratio approaches the boundary between "adjusting 2 layers" and "adjusting 3 layers," the mechanism prevents frequent switching between these two configurations, thereby avoiding redundant migrations and unnecessary costs. Moreover, in this boundary region, the performance difference between the two configurations is inherently minimal. Experimental results show that triggering redistribution only 5–7 times is sufficient to achieve an excellent balance between performance and stability.

**Memory Overhead Analysis.** To accommodate potential layer redistributions during runtime, we preemptively load the weights of layers that may be used in the future, which inevitably leads to a certain degree of GPU memory overhead. Taking the Qwen2.5-32B model as an example: under a static allocation strategy, each GPU is typically assigned 16 Transformer layers, whereas after dynamic redistribution, some GPUs may need to load up to 19 layers. For a 40GB A100 GPU, this

corresponds to approximately a 7.5% increase in memory usage. Although this adds some memory pressure, the resulting performance improvement is significant, reflecting a deliberate trade-off between system performance and memory resource utilization. More importantly, this memory overhead can be further optimized in the future. By introducing an asynchronous weight offloading and loading mechanism, the weights of layers that are not on the current computation path can be temporarily stored in CPU memory. When the scheduler anticipates an upcoming layer redistribution, the system asynchronously preloads the corresponding weights onto the target GPU in advance. This mechanism allows us to maintain high performance while significantly mitigating memory redundancy caused by dynamic layer redistribution.

## C  Performance Benefit Analysis.

Within the inference pipeline of LLM, the sampling stage typically introduces additional processing time after each microbatch's forward computation, resulting in pipeline bubbles. Under high concurrency, these bubbles accumulate continuously and significantly increase the queuing delay of subsequent requests. As illustrated in Fig. 1, in the baseline system, the accumulation of sampling bubbles progressively postpones the scheduling of subsequent microbatches into the pipeline. Let the duration of a single sampling bubble be $t_s$. If completing the prefill for the first $n$ requests requires $x$ microbatches, then the $(n+1)$-th request must wait at least $xt_s$ of additional time before it can start execution. Moreover, during decoding, each forward step introduces an extra $(\text{PP}_{\text{size}} - 1)$ sampling bubbles, further increasing the overall inference latency. In contrast, with layer redistribution optimization, bubble accumulation can be effectively mitigated, allowing new requests to enter the system more rapidly and reducing e2e latency.

The performance advantages of `DynaPipe` are most pronounced in scenarios with small models and high request rates. This phenomenon arises primarily because the per-step forward computation time of small models is short, whereas the sampling latency remains largely constant. Consequently, the relative contribution of sampling to total latency increases, thereby magnifying the effect of pipeline idle time on overall system performance.

