# OpenReview forum: "DynaPipe: Dynamic Layer Redistribution for Efficient Serving of LLMs with Pipeline Parallelism"
_NeurIPS.cc/2025/Conference — NeurIPS 2025 poster_

### Official Review · Reviewer_oJc6 · 2025-06-17

**Clarity:** 3
**Significance:** 2
**Originality:** 2
**Rating:** 4
**Confidence:** 3

**Summary:**

DynaPipe is an LLM serving framework that dynamically balances the workload across pipeline stages, by migrating KV cache at background. The evaluation observes significant advantages in performance and SLO guarantee.

**Questions:**

1. Can the authors quantify the impacts of post-processing on serving performance? or explain what are other factors lead to such huge (up to 49%) performance benefits? It seems that getting such benefits by just addressing the pipeline bubble caused by post-processing is too good to be true.

2. Is the work compatible with novel pipeline execution patterns (e.g., DualPipe)?

**Ethical Concerns:**

["NO or VERY MINOR ethics concerns only"]

**Final Justification:**

All my key concerns are solved in the rebuttal, but the evaluation benchmarks can still be improved (see limitation). The paper is solid, but addressing pipeline bubbles is not an entirely novel idea. Still, the good parts outweigh the parts that need improvement, so I am still positive about this paper.

**Limitations:**

The evaluation dataset is a bit limited (only on 2 smaller variants of Qwen). Maybe the authors can discuss more on what LLMs (e.g., small/large ones, MoE/dense ones, ...) are benefited more by DynaPipe

**Paper Formatting Concerns:**

no concern

**Quality:**

3

**Strengths And Weaknesses:**

Strengths

Pipeline bubbles caused by post-processing latency are a novelly identified issue.

The results (up to 49% performance improvement) seem too impressive.

Weakness

While bubbles caused by post-processing are a new issue, it is hard to see why addressing this small overhead can lead to such huge performance benefits.

---

> ### Author Rebuttal · Authors · 2025-07-31
>
> Thank you for your valuable comments. We appreciate your feedback and would like to address your concerns as follows.
>
> **Q1：Can the authors quantify the impacts of post-processing on serving performance? or explain what are other factors lead to such huge (up to 49%) performance benefits? It seems that getting such benefits by just addressing the pipeline bubble caused by post-processing is too good to be true.**
>
> First, we have included complete source code in the submitted package, and all experimental results are fully reproducible.
>
> Regarding the observed performance improvements, our analysis is as follows:
>
> First, the pipeline bubbles caused by sampling overhead tend to accumulate over time and significantly impact the waiting time of subsequent requests. As shown in the upper part of Figure 1, in the baseline system, these bubbles gradually accumulate, causing a noticeable delay in when a given microbatch enters the system compared to the optimized pipeline adopted by DynaPipe. This delay depends on the extent of bubble accumulation—the longer the accumulation time, the greater the delay.
>
> For example, suppose that at time $t_1$, the $(n+1)$-th request arrives. In DynaPipe, since the prefill phase for the previous $n$ requests has just completed, the new request can be seamlessly inserted into the system with virtually zero queuing delay. However, in the baseline system, if the prefill for the previous $n$ requests has not yet finished, and no computation resources are available, the new request must wait until the corresponding microbatch slot becomes available. Thus, the queuing time for this request equals the cumulative delay caused by all preceding sampling bubbles.
>
> Assume that it takes $n/2$ microbatches to complete the prefill for the previous $n$ requests, and that each microbatch execution introduces one sampling bubble with duration $t_s$. Then, the $(n+1)$-th request must wait at least $n/2 \times t_s$ before it can begin execution. Furthermore, once the request starts decoding, each forward computation step still suffers from an additional $(pp\_{size} - 1)$ sampling bubbles compared to DynaPipe, as illustrated in Figure 1. Therefore, both the queuing latency and the effective throughput are negatively impacted by the inter-stage idle time introduced by sampling.
>
> As for why we observe performance gains as high as 49%, we explain this as follows:
>
> The best-case improvement occurs when using the Qwen2.5-14B model at a request rate of 36. In this scenario, the model is relatively small, resulting in lower per-layer forward computation cost, whereas the sampling overhead remains the same. As a result, the ratio of sampling overhead to forward computation cost increases, making the impact of sampling more pronounced.
>
> Moreover, under high request rates (e.g., 36), the incoming requests quickly accumulate in the queue. In such cases, the queuing delay of at least $n/2 \times t_s$ can become substantial. For requests with short output lengths, the queuing time may even exceed the actual inference time, drastically reducing system efficiency.
>
> Therefore, DynaPipe's optimization significantly mitigates the idle time caused by sampling bubbles and improves resource utilization during forward computation. This leads to up to 49% end-to-end performance improvement under specific but practically relevant settings.
>
> **Q2: Is the work compatible with novel pipeline execution patterns (e.g., DualPipe)?**
>
> DualPipe is an optimization method designed for large language model training, while our proposed DynaPipe focuses on performance optimization during the inference stage. This distinction arises because training involves not only forward computation but also backpropagation and parameter updates. If dynamic layer redistribution is applied during training, it may disrupt the alignment between the backward pass and the corresponding parameter layers, leading to incorrect or failed parameter updates. Therefore, DynaPipe's dynamic redistribution mechanism is not suitable for training scenarios. Its strengths lie in the inference stage, where it can dynamically adapt to runtime workload fluctuations and improve overall execution efficiency.
>
> **Response to Limitation:**
>
> We acknowledge that the current evaluation is limited to two relatively small variants of the Qwen model, and the dataset scope is somewhat narrow. However, from a mechanism perspective, DynaPipe is generally applicable to LLMs of varying sizes and architectures.
>
> Specifically, the overhead of the sample phase mainly depends on the vocabulary size and is largely independent of the model parameter scale. In contrast, the forward computation cost of each layer is closely tied to the model’s weight parameters. As a result, DynaPipe tends to yield more significant benefits for smaller models, while its effectiveness may decrease as model size grows. That said, in practical deployments, especially on edge devices, many use mid-sized models such as 14B or 32B, and our experiments demonstrate that DynaPipe achieves substantial performance gains on these models.
>
> Moreover, DynaPipe is effective for both dense and sparse (MoE) models. Our current experiments are based on dense models. In the case of MoE models, although the expert selection mechanism may lead to load imbalance, pipeline parallelism typically assigns all experts within a layer to the same GPU, thereby avoiding inter-expert imbalance. However, the sample operation is still executed on the final GPU stage, which introduces inter-stage pipeline bubbles. In such scenarios, DynaPipe can still be leveraged to mitigate the performance degradation through dynamic layer redistribution.

---

> > ### Comment · Reviewer_oJc6 · 2025-08-01
> >
> > Q1: Thanks for the explanation! Then the question is, how common is the case "high request rate + low output length" in the real world? I guess it is not true for the large LLMs used for CoT or agentic applications, but it is probably true for small LLMs on edge devices, so I am wondering if there is any way to confirm that? I think this point will make your motivation stronger.
> >
> > Q2: Answer addressed my question.
> >
> > Q3: Even though you can only access a limited set of LLMs and all LLMs have similar architectures, I still recommend having at least 3 different-sized models in the evaluation. Using two models to demonstrate the trend "small models get more benefits" is a bit insufficient. However, since I should not ask for any new data here, this is just a recommendation you may consider during a future revision.

---

> > > ### Author Response · Authors · 2025-08-02
> > >
> > > Thank you for your thoughtful feedback. We truly appreciate your suggestions. Below are our responses to your comments:
> > >
> > > **Q1:** We apologize for not fully clarifying this point in our initial reply. Previously, we only mentioned one scenario, namely high request rate combined with low output length, but that does not mean our method performs poorly when the output length is longer. As shown in Figure 4, the output-to-input length ratio in the ShareGPT dataset is higher than that of the Azure-Conv dataset, and our method achieves even stronger improvements on ShareGPT. Furthermore, Figure 5 shows that when output lengths increase, latency grows significantly if redistribution is not applied.
> > >
> > > This is because, in datasets with longer average output lengths, each request involves more decoding steps. Under the chunked prefill mechanism, this leads to more decode requests being accumulated within each batch. As the number of decode requests increases, the overall sampling overhead also increases. As a result, pipeline bubbles caused by sampling overhead become more severe.
> > >
> > > In addition, for tasks such as CoT that produce longer outputs, each request takes longer to complete, which means the corresponding KV cache is held for a longer time. When GPU memory is limited, this delays the admission of new requests into the system. Combined with the bubble accumulation effect caused by sampling overhead, this further increases the end-to-end latency.
> > >
> > > Regarding different request rates, we have demonstrated in Figure 4 that our approach provides optimization benefits across varying levels of load, not just under high request rates.
> > >
> > > **Q3:** Thank you for this suggestion. Due to space limitations, we did not include multiple models in the current version of the paper, which we acknowledge as a limitation. We will consider adding more experimental results in future revisions to better support our conclusions.
> > >
> > > That said, we believe our reasoning is theoretically sound. Our method does not modify model architectures such as LLaMA, Qwen, or ChatGLM. Instead, it adjusts the placement of layers across GPUs at a per-layer granularity. Since the sampling operation is a common step across all decoder-only transformer models, our approach should be applicable to a wide range of models.
> > >
> > > Regarding model size, as we mentioned in our previous response under the limitations discussion, Figure 4 shows that smaller models tend to benefit more from our approach. This is because the sampling overhead, which is mainly determined by vocabulary size, constitutes a larger portion of total computation in smaller models. Nevertheless, your point is well taken. To ensure completeness, we will include a more comprehensive evaluation in future revisions.
> > >
> > > Thank you again for your valuable comments. Please feel free to reach out if you have any further questions or suggestions.

---

> > > > ### Comment · Reviewer_oJc6 · 2025-08-02
> > > >
> > > > Thanks for your reply!
> > > > Q1: the answer fully addresses my concern.
> > > > Q3: also good for now.

---

> > > > > ### Author Response · Authors · 2025-08-06
> > > > >
> > > > > Thank you very much once again for your valuable feedback! We will revise the paper based on your suggestions and supplement it with additional experiments to strengthen its persuasiveness. Meanwhile, we would like to kindly ask whether you would consider adjusting your score, or if our responses have helped improve your assessment of our work. If you have any further questions or suggestions, we would be more than happy to continue the discussion.

---

### Official Review · Reviewer_oYmA · 2025-06-24

**Clarity:** 3
**Significance:** 2
**Originality:** 2
**Rating:** 4
**Confidence:** 4

**Summary:**

This paper focuses on a significant but overlooked bottleneck in PP caused by sampling operations during LLM inference. It proposes a dynamic layer redistribution approach and an asynchronous KV cache migration technique to address this bottleneck. Experiments show that the proposed method can reduce the average latency and improve SLO attainment rate compared to existing systems.

**Questions:**

* The same problem has been well studied for LLM training in [1], where they propose to partition the sampling operations across PP devices and adapt PP scheduling. Compared to trivial layer redistribution, this method provides a unified solution under all configurations, with quite small overhead and balanced memory. Can similar ideas be applied in LLM inference as a better solution (to resolve the limitations mentioned in Weaknesses)? BTW, I would suggest to cite papers on this topic for LLM training  ([1] and potentially more) in Related Work.
* From my understanding, the batch in an inference service is dynamic, especially for online inference where new requests can come at any time. How to guarantee the optimal solution under such dynamic settings?

[1] Yeung, M. T., Qi, P., Lin, M., and Wan, X. Balancing pipeline parallelism with vocabulary parallelism. arXiv preprint arXiv:2411.05288, 2024.

**Ethical Concerns:**

["NO or VERY MINOR ethics concerns only"]

**Final Justification:**

The topic is reasonable and the engineering effort is commendable. I am open to raising my score from 3 to 4, but only after the following points are addressed:
* The authors should clearly state the limitations; a dedicated limitations section is encouraged.
* There should be a thorough discussion of existing solutions for both training and inference to help readers better understand the broader context and alternative approaches.

**Quality:**

2

**Strengths And Weaknesses:**

## Strengths
* The target problem is valid and underexplored.
* The paper is well written.
* The evaluation is extensive and detailed.

## Weaknesses
* The proposed method introduces migration overhead (both parameters and KV cache), and this overhead can be huge in some configurations (totally different layers for some stages).
* The paper lacks memory analyses. For example, layer redistribution will make KV cache imbalanced and may introduce memory overhead.
* For many cofigurations, there still exists computational imbalance becuase layer redistribution cannot provide perfect solutions for all use cases.

---

> ### Author Rebuttal · Authors · 2025-07-31
>
> Thank you for your valuable comments. We appreciate your feedback and would like to address your concerns as follows.
>
> **Q1: The same problem has been well studied for LLM training in [1], where they propose to partition the sampling operations across PP devices and adapt PP scheduling. Compared to trivial layer redistribution, this method provides a unified solution under all configurations, with quite small overhead and balanced memory. Can similar ideas be applied in LLM inference as a better solution (to resolve the limitations mentioned in Weaknesses)? BTW, I would suggest to cite papers on this topic for LLM training ([1] and potentially more) in Related Work.**
>
> We will cite the paper *Balancing Pipeline Parallelism with Vocabulary Parallelism* in the related work section. However, we believe the technique proposed in that paper is primarily designed for LLM training scenarios and has certain limitations when applied to inference workloads.
>
> The paper proposes a vocabulary parallelism approach that distributes the sampling operation, which is originally concentrated in the final pipeline stage, across all GPUs. This effectively amortizes the sampling overhead over all pipeline stages. While this method is effective during training, it largely benefits from the presence of backward propagation dependencies. These dependencies prevent intermediate pipeline stages from continuously executing forward passes, requiring them to wait for backward computations on certain microbatches to complete. As a result, idle periods ("bubbles") naturally arise in the pipeline, which can be utilized to perform sampling operations and improve resource efficiency.
>
> However, this assumption no longer holds in inference scenarios. First, since there is no backward propagation, each pipeline stage can continuously process forward passes of its assigned microbatches. Second, pipeline parallelism allows at most `pp_size` microbatches to be processed concurrently in the system. When the microbatch at the final stage has not completed its sampling operation, a new microbatch cannot be injected into the pipeline, causing the first stage to be blocked and generating front-end bubbles. Therefore, in inference, sampling operations must be completed as early as possible to ensure continuous and efficient pipeline execution.
>
> Introducing vocabulary-parallel sampling during inference also poses challenges. Coordinating the sampling operation across all pipeline stages requires careful synchronization. In practice, due to the variability in forward pass durations across pipeline stages, such synchronization may actually delay the sampling completion. For example, suppose GPU3 finishes the forward pass of microbatch 0 and broadcasts a signal to all GPUs to perform the sampling operation. At that moment, GPU2 may have just started executing microbatch 3. It must finish the current forward pass before executing the sampling, resulting in a sampling delay of at least one microbatch's execution time. This delay would hinder the injection of new microbatches into the pipeline and degrade overall throughput.
>
> Furthermore, the vocabulary-parallel mechanism divides the output layer computation into a local computation phase (S) and a global normalization phase (T), where S must occur after the final Transformer layer, and T must wait for all S phases to complete. This introduces additional global communication and synchronization, which is particularly undesirable in inference workloads. It increases scheduling complexity, creates new performance bottlenecks, and introduces pipeline bubbles.
>
> In contrast, our proposed Dynamic Layer Redistribution technique achieves load balancing without incurring the cost of global synchronization and communication, enabling more efficient pipeline-parallel execution for inference scenarios.
>
> **Q2: From my understanding, the batch in an inference service is dynamic, especially for online inference where new requests can come at any time. How to guarantee the optimal solution under such dynamic settings?**
>
> Indeed, in online inference services, the batch of incoming requests is inherently dynamic. In our experiments, we simulate this scenario by controlling the request arrival rate to mimic an online serving environment. Because requests arrive at different time points, the ratio of prefill to decode requests within each microbatch varies over time.
>
> We adopt a chunked prefill mechanism, which allows prefill and decode requests to be mixed and executed together within a single microbatch. Under this mechanism, the execution time of each Transformer layer depends on the total number of tokens (prefill + decode), while the sampling overhead only depends on the number of decode requests. As a result, fluctuations in the prefill/decode ratio, driven by the dynamic arrival of new requests, lead to corresponding changes in the relative cost between sampling and single-layer forward computation.
>
> In such a dynamic setting, static layer partitioning becomes suboptimal, as it cannot adapt to the shifting workload composition, often leading to pipeline bubbles and resource imbalance. To address this, we propose a dynamic layer redistribution mechanism that adjusts to real-time workload variations, effectively minimizing pipeline stalls and improving overall system throughput.
>
> **Response to Weaknesses:**
>
> （1）**Regarding the migration overhead**, the number of KV Cache layers that need to be transferred varies across different pipeline stages. As a result, in certain cases, the last GPU may incur significantly more migration traffic than the first GPU, introducing additional cost.
>
> To mitigate this overhead, we adopt two key optimization strategies:
>
> 1. Asynchronous communication-computation overlap: By overlapping KV cache migration with partial forward computation, we are able to hide part of the communication latency.
> 2. Window threshold mechanism (see Section 4.4): Instead of triggering a layer redistribution every time the ratio between sampling cost and per-layer forward cost changes, we accumulate this signal over a sliding window. Only when it reaches a predefined threshold do we trigger a reallocation. This approach prevents frequent layer adjustments caused by transient or local fluctuations, thereby reducing unnecessary migration overhead.
>
> As shown in Figure 6, when the window threshold is set to 25, the system achieves a good trade-off between performance gain and overhead, effectively suppressing the cost introduced by frequent layer reassignments.
>
> In fact, the primary motivation for introducing the window threshold is to avoid frequent oscillation between boundary configurations. For example, when the sampling-to-forward ratio hovers near the decision boundary between adjusting 2 layers and adjusting 3 layers, the system may repeatedly switch back and forth—adjusting to 3 layers in one round and reverting to 2 layers in the next. Such frequent changes bring little performance benefit but lead to excessive KV cache migrations, increasing overhead instead.
>
> More importantly, in this boundary range, the performance difference between adjusting 2 layers and adjusting 3 layers is negligible. Therefore, the window mechanism helps suppress unnecessary jitter, ensuring stable performance improvements while significantly reducing the cost associated with frequent KV cache movements.
>
> （2）**Regarding memory overhead**, dynamic layer redistribution does introduce additional GPU memory usage. However, from a system-wide perspective, this represents a trade-off between performance and memory consumption, as redistributing layers helps eliminate pipeline bubbles and improves overall throughput.
>
> Take Qwen-32B as an example: originally, each GPU holds around 15 transformer layers. After redistribution, this increases to 18 layers, resulting in approximately 16% more GPU memory usage. However, this overhead can be further optimized and mitigated.
>
> We are currently implementing an enhancement that caches the surplus weights in CPU memory. When the scheduler detects the need for a layer redistribution, it preloads the required weights asynchronously from CPU to GPU, just before they are needed. This allows us to keep the number of resident weights on the GPU low without sacrificing performance, thus significantly reducing redundant memory usage.
>
>
> （3）**Generality**: While there are still other types of computational load imbalance in the system, this paper focuses specifically on the performance bottleneck caused by sampling overhead. We argue that this issue has a significant impact during the inference stage and thus deserves focused attention and targeted solutions.
>
> It is also worth noting that our proposed approach is orthogonal to other load-balancing techniques, meaning it can be integrated with existing solutions to jointly address broader load imbalance challenges and further improve overall inference efficiency.

---

> > ### Comment · Reviewer_oYmA · 2025-08-01
> >
> > Firstly, I do agree that Vocabulary Parallelism for inference system faces some different technical challanges, for example, how to efficiently implement top-p&top-k as in FlashInfer, and how to reduce synchronization. In the original Vocabulary Parallelism paper (for training), I believe they use asynchronous communication to overlap with computation, and insert vocabulary computations into the scheduling. So the argument of `creates new performance bottlenecks, and introduces pipeline bubbles` is not convincing enough.
> >
> > BTW, from my understanding, most of the imbalance comes from the last linear layer (computing logits), instead of the sampling operation?
> >
> > > We argue that this issue has a significant impact during the inference stage and thus deserves focused attention and targeted solutions.
> >
> > I do agree that it is a valuable topic, and the engineering effort deserves respect. However, from my perspective, at least two concerns should be addressed in the paper:
> > * The authors should clearly state the limitations, such as memory overhead, migration overhead, potentially suboptimal solutions due to the dynamic nature of the approach, and that imbalance may still occur in many scenarios. A dedicated limitations section is encouraged.
> > * There should be a thorough discussion of existing solutions for both training and inference. This will help readers better understand the broader context and alternative approaches.
> >
> > Finally, I understand it may not be realistic to develop Vocabulary Parallelism (or other potential solutions) within the scope of this paper. However, I suggest the authors consider some exploration in this direction, as the limitations mentioned above are inherent to layer redistribution.

---

> > > ### Author Response · Authors · 2025-08-02
> > >
> > > Thank you for your positive response and valuable suggestions, which have been truly inspiring for us. Below are our replies to your comments:
> > >
> > > 1. **On the Limitations of Vocabulary Parallelism in Inference**
> > >
> > >    We do not deny that the Vocabulary Parallelism paper leverages asynchronous communication to overlap computation. However, the inference scenario is fundamentally different from training. Training involves backward propagation, which naturally introduces pipeline bubbles that provide scheduling space for vocabulary computation. As shown in our Figure 1 and compared with Figure 10 in the Vocabulary Parallelism paper, inference can maintain a steady flow of microbatches as long as sampling is completed promptly. In contrast, training must wait for the backward pass to finish.
> > >
> > >    Therefore, in inference, **sampling must be completed as early as possible** to allow the system to move on to the next microbatch. Vocabulary parallelism faces challenges here: in online inference, microbatches are not uniform in size, and each GPU may be executing different microbatches at a given time. As we previously mentioned, even if the last GPU finishes its microbatch and signals others to start vocabulary parallelism, the remaining GPUs might still be processing their own microbatches or have already started new ones. In such cases, even though other GPUs could finish their respective sampling tasks earlier, the overall completion is gated by that one GPU, delaying the entry of the next microbatch. This prevents new microbatches from entering the pipeline, resulting in bubbles. This issue does not typically occur during training, where microbatch sizes can be controlled offline and backward propagation naturally provides scheduling slack to enable finer-grained coordination of sampling, as discussed in the original paper.
> > >
> > > 2. As noted in a footnote on the first page of our paper, we use the term "sampling" to collectively refer to both the logits computation and the sampling operation, for simplicity. Therefore, our method addresses the load imbalance issues introduced by both components. However, the Vocabulary Parallelism paper only addresses the sampling operation itself, without tackling the imbalance introduced by the logits computation.
> > >
> > > 3. **Limitations Discussion:**
> > >     We fully agree with your suggestion and will expand our analysis of limitations in the final version. Due to space constraints, we provide a brief summary here:
> > >
> > >    - While our method introduces some memory and communication overhead (approximately 15% additional memory usage), this is not a fundamental limitation. We are actively optimizing this by placing redundant model weights on the CPU and loading them asynchronously as needed, rather than occupying GPU memory from the start.
> > >    - Dynamic layer redistribution does introduce some migration overhead, but the cost is currently acceptable, as it leads to a significant reduction in end-to-end request latency — making it a worthwhile trade-off. Moreover, we control the adjustment frequency using a window-based threshold, which prevents excessive reassignments. In practice, only a few adjustments are needed to achieve substantial performance gains, and the overhead introduced is almost negligible in the overall execution.
> > >    - The main limitation of our work is that it only addresses inter-stage imbalance caused by sampling overhead. However, this issue is particularly critical in inference scenarios. Our method is orthogonal to other techniques that address different types of imbalance — for example, it can be combined with approaches such as chunk prefill.
> > >
> > > 4. **Related Work Discussion:**
> > >     We will extend our related work section accordingly. In general, sampling has received relatively little attention in inference scenarios. Solutions proposed for training are often ill-suited for inference due to the absence of backward computation. As a result, sampling latency becomes more prominent in inference and should not be overlooked.
> > >
> > > Thank you again for your constructive feedback. We will continue to refine our approach and work on reducing the overhead introduced by dynamic layer redistribution. We also plan to explore how the idea of vocabulary parallelism can be leveraged to distribute this operation across GPUs, in order to avoid introducing additional memory overhead.

---

> > > > ### Comment · Reviewer_oYmA · 2025-08-04
> > > >
> > > > >pipeline bubbles that provide scheduling space for vocabulary computation.
> > > >
> > > > >Vocabulary Parallelism paper only addresses the sampling operation itself, without tackling the imbalance introduced by the logits computation.
> > > >
> > > > These two arguments are not true. Vocabulary Parallelism does not utilize pipeline bubbles, and it also does the partition on logits computation.
> > > >
> > > >
> > > > > inference can maintain a steady flow of microbatches as long as sampling is completed promptly. In contrast, training must wait for the backward pass to finish.
> > > >
> > > > There is also a bubble-free stable phase for PP training, which is known as one-forward-one-backward. I don't understand why waiting for backward causes fundamental differences.
> > > >
> > > > ----------------------------
> > > >
> > > > As I'm only asking for a clear limitation statement and a thorough discussion of existing solutions, futher discussion on Vocabulary Parallelism does not help too much on improving this paper. Again, I would encourage the authors to further explore this direction in their future work.

---

> > > > > ### Author Response · Authors · 2025-08-04
> > > > >
> > > > > Thank you very much for your kind reply. We highly appreciate and fully recognize your comments, and we will continue to improve the paper accordingly in future revisions.
> > > > >
> > > > > In the current version, we have already provided a relatively detailed analysis of the limitations of our method. Regarding the discussion of existing solutions, we have observed that this issue has received very limited attention in the inference setting. Most related works are focused on the training stage, such as those employing static layer allocation (e.g., Section B.2 in *Skywork-MoE* [1]) or approaches based on vocabulary parallelism. These works have also acknowledged the performance bottleneck caused by inter-stage imbalance introduced by the logits computation and sampling stages, which, in turn, validates the importance and correctness of the specific problem we aim to address.
> > > > >
> > > > > As for the soundness of our proposed design, we agree that more experimental evidence would be helpful. We have already compared against static redistribution schemes in the current paper. For vocabulary parallelism, we plan to further investigate its implementation details and explore how to adapt it for inference scenarios, so that we can conduct empirical comparisons and enrich the experimental evidence of our approach. Nevertheless, we believe our method demonstrates a degree of novelty and effectiveness in addressing scheduling and system-level inefficiencies in current LLM inference workloads.
> > > > >
> > > > > We also acknowledge that vocabulary parallelism is an elegant and promising scheduling strategy. After adapting it to the inference system, we intend to analyze and benchmark it more thoroughly, and explore how to integrate its strengths to further improve our design.
> > > > >
> > > > > Considering these aspects, we respectfully and kindly invite you to re-evaluate the rating of our submission. We greatly appreciate your valuable feedback and look forward to any further comments or suggestions you may have.
> > > > >
> > > > > [1] Skywork-MoE: A Deep Dive into Training Techniques for Mixture-of-Experts Language Models.

---

> > > > > > ### Comment · Reviewer_oYmA · 2025-08-06
> > > > > > **Final Justification**
> > > > > >
> > > > > > **Final Justification**
> > > > > >
> > > > > > The topic is reasonable and the engineering effort is commendable. I am open to raising my score from 3 to 4, but only after the following points are addressed:
> > > > > >
> > > > > > * The authors should clearly state the limitations; a dedicated limitations section is encouraged.
> > > > > > * There should be a thorough discussion of existing solutions for both training and inference to help readers better understand the broader context and alternative approaches.

---

> > > > > > > ### Author Response · Authors · 2025-08-06
> > > > > > >
> > > > > > > Thank you for your suggestion. Based on your feedback, we have revised the following two subsections accordingly.
> > > > > > >
> > > > > > > **Limitation**: This work makes several design trade-offs that may limit its applicability in larger-scale or more complex deployment scenarios. The proposed dynamic layer redistribution mechanism requires each GPU to load additional model weights, resulting in approximately a 10% reduction in memory space available for KV cache allocation. In addition, the runtime KV cache migration introduces communication overhead, which may lead to temporary performance fluctuations during redistribution. Future work may mitigate these issues by asynchronously preloading weights on the CPU to reduce GPU memory pressure, and by adopting finer-grained KV cache transfer strategies to better overlap communication and computation.
> > > > > > >
> > > > > > > Moreover, this work primarily focuses on pipeline parallelism, without exploring the joint optimization of other mainstream parallelism strategies such as tensor parallelism, sequence parallelism, and expert parallelism. In practice, these strategies are often used in combination, introducing more complex scheduling and coordination challenges. Extending DynaPipe to support multi-dimensional parallelism is an important direction for future research.
> > > > > > >
> > > > > > > **Related Work**
> > > > > > >  (This section complements existing related work in the paper by specifically addressing the issue of inter-stage imbalance.)
> > > > > > >
> > > > > > > In recent years, works such as SARATHI [1] and gLLM [2] have explored pipeline parallelism for inference, aiming to reduce pipeline bubbles through stage parallelism and load balancing. TD-Pipe [3] further proposes a temporally disaggregated pipeline architecture to better coordinate the prefill and decode phases. However, these approaches overlook the inter-stage imbalance caused by the sampling overhead, which can lead to underutilization of GPU resources. Orthogonal to these efforts, we propose a dynamic layer redistribution mechanism to address this imbalance and improve inference efficiency.
> > > > > > >
> > > > > > > In the context of LLM training, prior work has primarily focused on communication latency [4], memory balancing [5, 6, 7], pipeline bubble mitigation [6, 8, 10, 11], and activation checkpointing [5, 9]. Some studies have noted inter-stage imbalance; for example, Skywork-MoE [11] and Megatron-LM [12] adopt static layer redistribution to alleviate this issue. However, such static approaches are less effective under the dynamic workload conditions typical of model serving. In contrast, [10] proposes a novel Vocabulary Parallelism method to balance compute and memory in pipeline parallelism, but it is designed specifically for training scenarios and may incur high communication overhead and synchronization complexity when applied to inference. Our proposed DynaPipe dynamically adjusts layer distribution based on real-time workload conditions and only triggers redistribution when significant changes occur, resulting in relatively low communication overhead suitable for inference serving.
> > > > > > >
> > > > > > > Thank you for your time and effort in reviewing our paper. Please feel free to reach out if you have any further questions or suggestions.
> > > > > > >
> > > > > > > [1] SARATHI: efficient LLM inference by piggybacking decodes with chunked prefills
> > > > > > >
> > > > > > > [2] gllm: Global balanced pipeline parallelism system for distributed llm serving with token throttling
> > > > > > >
> > > > > > > [3] TD-Pipe: Temporally-Disaggregated Pipeline Parallelism Architecture for High-Throughput LLM Inference
> > > > > > >
> > > > > > > [4] Weipipe: Weight pipeline parallelism for communication-effective long-context large model training
> > > > > > >
> > > > > > > [5] Adapipe: Optimizing pipeline parallelism with adaptive recomputation and partitioning
> > > > > > >
> > > > > > > [6] Hanayo: Harnessing wave-like pipeline parallelism for enhanced large model training efficiency
> > > > > > >
> > > > > > > [7] Bpipe: Memory-balanced pipeline parallelism for training large language models
> > > > > > >
> > > > > > > [8] Zero bubble (almost) pipeline parallelism
> > > > > > >
> > > > > > > [9] Mario: Near zero-cost activation checkpointing in pipeline parallelism
> > > > > > >
> > > > > > > [10] BALANCING PIPELINE PARALLELISM WITH VOCABULARY PARALLELISM
> > > > > > >
> > > > > > > [11] Skywork-MoE: A Deep Dive into Training Techniques for Mixture-of-Experts Language Models.
> > > > > > >
> > > > > > > [12] Megatron-LM: Training Multi-Billion Parameter Language Models Using Model Parallelism

---

### Official Review · Reviewer_Zxmv · 2025-06-28

**Clarity:** 3
**Significance:** 2
**Originality:** 2
**Rating:** 4
**Confidence:** 4

**Summary:**

This paper proposes DynaPipe, a system that optimizes pipeline parallel inference by dynamically redistributing layers between stages. The authors observe the sampling (occurring at the last stage of the pipeline) causes stage imbalance. The authors propose to reduce the number of layers in the last stage (and redistribute them to previous stages) and improve pipeline balance. The optimal redistribution is determined by a performance model, while asynchronous pre-transferring of KV cache is employed to reduce redistribution overheads.

**Questions:**

— In Fig.2, it seems both sampling and layer computation time grows near linearly with the number of tokens. What is the source of the variance of sample / layer time (Fig. 2b)? Is it because sampling and layer time grow at a different rate?

— What is the model used in Fig.2? Is the sampling time mainly related to vocabulary size (and the number of tokens to be decoded), rather than model size? If so, will the sampling time become negligible if the model is large enough?

— How does the sample-layer time ratio change during real serving scenarios? The claim that dynamic adjustments are needed can be better illustrated if authors can show the variance of sample-layer time ratio over a time interval, under realistic settings.

—The paper assumes chunked prefill is used. However, production inference services often adopt disaggregated serving (e.g., [3]), where prefill and decoding happen on different GPUs. Does the analysis and the approach proposed in this paper still apply?

—In Sec.3.3, it is stated that "the system pre-allocates memory and loads the weights for additional layers that may be reassigned". How many layers are loaded in advance? Will this cause extra memory consumption?

—In Sec.4.3, around line 281-282, why output length is related to sampling overhead? Is it because it takes more time to sample each token, or it's because there's more decoding tokens (rather than prefill)?

—What is the layer partitioning strategy used in the gLLM baseline? While static partitioning is used as baselines in Fig.5, it would be nice if static partitioning (with uneven layer distribution) is compared to in end-to-end evaluation on real datasets, to better justify the need for dynamic adjustments.

[3] https://arxiv.org/pdf/2401.09670

**Ethical Concerns:**

["NO or VERY MINOR ethics concerns only"]

**Final Justification:**

The authors have addressed my major concerns satisfactorily.

**Limitations:**

Yes

**Paper Formatting Concerns:**

Nil

**Quality:**

2

**Strengths And Weaknesses:**

Strengths:
 1. The paper makes an interesting observation that the sampling overhead can cause pipeline imbalance during inference.
 2. The methods proposed in the paper are simple and easy to implement.

Weakness:

— The problem of pipeline imbalance when layers are uniformly partitioned is well known in training, often mitigated by partitioning the layers unevenly (e.g., sec. B.2 in Skywork-MoE[1], configuration option in Megatron-LM [2]). Other optimizations like asynchronous transfer of KV Cache are also standard techniques. Therefore, it would be better if the paper focuses more on justifying the need for dynamic justification of layers (which differentiates the paper from prior works).

—It would be better to quantify the overhead of dynamic layer & KV cache redistribution in the evaluation section.

[1] https://arxiv.org/pdf/2406.06563

[2] https://github.com/NVIDIA/Megatron-LM/blob/45400df7da7fa23e3aff86804e5ac254d9a8d3c0/megatron/training/arguments.py#L2258

---

> ### Author Rebuttal · Authors · 2025-07-31
>
> Thank you for your valuable comments. We appreciate your feedback and would like to address your concerns as follows.
>
> **Q1:In Fig.2, it seems both sampling and layer computation time grows near linearly with the number of tokens. What is the source of the variance of sample / layer time (Fig. 2b)? Is it because sampling and layer time grow at a different rate?**
>
> Figure 2(a) illustrates that the sampling overhead grows linearly with the number of sampled tokens. The number of sampled tokens is determined by the number of decode requests in a batch, since only decode requests require sampling operations. Let the sampling overhead be denoted as $t_s$, and the number of decode tokens as $d_n$, then: $t_s = a \cdot d_n$.
>
> Figure 2(c) shows that the forward execution time of a single Transformer layer increases with the total number of tokens to be processed, including both prefill and decode tokens. Theoretically, due to the quadratic complexity of the attention mechanism, the forward time should grow quadratically with respect to the token count. However, in practice, the attention component contributes only a small portion of the total execution time, with the majority dominated by MLP and other linear computations. As a result, the forward time empirically appears to scale linearly.
>
> Thus, we simplify the forward cost per layer as a linear function. Let the forward time per layer be $f_t$, and the number of prefill tokens be $p_n$, then:$t_f = b \cdot (p_n + d_n)$.
>
> Consequently, the ratio between sampling overhead and forward computation time becomes:$\frac{t_s}{t_f} = \frac{a}{b} \cdot \left( \frac{p_n}{d_n} + 1 \right)^{-1}$
>
> In the chunked prefill setting, prefill and decode requests are mixed and processed jointly within a single batch. Since the number of decode and prefill tokens varies dynamically across batches due to changing workloads, the ratio $p_n / d_n$ also fluctuates over time. This leads to a dynamic change in the ratio between sampling and forward computation costs.
>
> To address this challenge and adapt to the evolving workload, we introduce our dynamic layer redistribution design, which enables more efficient pipeline parallelism under mixed prefill/decode batch processing.
>
> **Q2:What is the model used in Fig.2? Is the sampling time mainly related to vocabulary size (and the number of tokens to be decoded), rather than model size? If so, will the sampling time become negligible if the model is large enough?**
>
> Your observation is absolutely correct: the sampling overhead is primarily determined by the vocabulary size and the number of tokens to be decoded, and is largely independent of the model's parameter scale. In very large models such as DeepSeek 671B, the forward computation dominates the total runtime, making the sampling cost relatively negligible.
>
> However, as shown in Figure 2, our experiments are conducted on the Qwen2.5-32B model, where the sampling overhead is already significant. More importantly, in many real-world deployment scenarios, particularly in on-device applications, extremely large models are not commonly used. Models such as 32B and 72B are widely adopted in production because they offer a favorable trade-off between performance and efficiency.
>
> At these model scales, sampling can become a performance bottleneck. Therefore, addressing the sampling overhead remains practically important.
>
> **Q3：How does the sample-layer time ratio change during real serving scenarios? The claim that dynamic adjustments are needed can be better illustrated if authors can show the variance of sample-layer time ratio over a time interval, under realistic settings.**
>
> As we discussed in our response to Q1, the ratio between sampling time and per-layer computation time dynamically changes according to the proportion of prefill and decode tokens within each batch.
>
> We will include a visualization of this ratio’s temporal variation in the final version of the paper.    Here, we provide a textual description of its general trend in real-world serving scenarios:
>
> In the early stage, batches are dominated by prefill requests, with relatively few decode tokens.    As a result, the sampling overhead is small compared to the forward computation time.
>
> As decoding progresses, the number of decode requests increases, causing the sampling overhead to constitute an increasingly larger proportion relative to the forward computation.
>
> In the middle stage, due to the dynamic arrival of new requests in the serving environment, the number of prefill tokens fluctuates, causing the ratio to change dynamically.
>
> In the final stage, as the batch consists solely of decode requests, the ratio reaches its maximum.
>
> **Q4：The paper assumes chunked prefill is used. However, production inference services often adopt disaggregated serving (e.g., [3]), where prefill and decoding happen on different GPUs. Does the analysis and the approach proposed in this paper still apply?**
>
> We believe that the proposed method remains applicable — and even more critical — in serving architectures where prefill and decode are decoupled. This is because sampling overhead only occurs during the decode stage, allowing for targeted optimization of the decode cluster. Since the decode cluster no longer performs forward computation for prefill tokens, the relative cost of sampling becomes even more prominent, potentially turning into a significant performance bottleneck.
>
> Moreover, in real-world scenarios, especially when serving long text inputs, the computational cost of attention increases substantially. In such cases, the forward computation time exhibits a nonlinear, approximately quadratic relationship with the number of tokens. A simplified model would express the forward time as:$t_f = b \cdot d_n^2$, while the sampling time remains:$t_s = a \cdot d_n$, resulting in a ratio of:$\frac{t_s}{t_f} = \frac{a}{b} \cdot \frac{1}{d_n}$.
>
> This indicates that the ratio of sampling to forward computation cost dynamically varies with the number of decode tokens. Therefore, even under disaggregated serving architectures, our proposed dynamic layer redistribution strategy remains essential for adapting to workload variations and optimizing overall inference performance.
>
> **Q5:In Sec.3.3, it is stated that "the system pre-allocates memory and loads the weights for additional layers that may be reassigned". How many layers are loaded in advance? Will this cause extra memory consumption?**
>
> Our current strategy preloads on each GPU the model weights that may be needed in the future. For example, in a 4-GPU setting, up to three layers may be dynamically reassigned, so each GPU loads an additional three layers of weights in advance. This strategy does introduce some extra GPU memory overhead, but it represents a trade-off between performance and resource consumption. Based on our experimental results, this approach significantly reduces end-to-end latency for individual requests.
>
> A similar strategy is also adopted in the deployment of DeepSeek, where redundant expert models are loaded on each GPU to improve overall system efficiency.
>
> Moreover, we are actively developing an optimization to reduce this additional GPU memory usage. Specifically, the redundant weights are preloaded into CPU memory. During runtime, a predictor anticipates potential layer reassignments and asynchronously transfers the required weights to the GPU while offloading the weights of unused layers. This mechanism retains the flexibility of dynamic scheduling while effectively lowering GPU memory consumption.
>
> **Q6:In Sec.4.3, around line 281-282, why output length is related to sampling overhead? Is it because it takes more time to sample each token, or it's because there's more decoding tokens (rather than prefill)?**
>
> This is because, in a dataset with a longer average output length, each request undergoes more decoding steps. Under the chunked prefill mechanism, this leads to a higher number of decode requests accumulated within each batch. As the number of decode requests increases, the overall sampling overhead correspondingly grows.
>
> **Q7:What is the layer partitioning strategy used in the gLLM baseline? While static partitioning is used as baselines in Fig.5, it would be nice if static partitioning (with uneven layer distribution) is compared to in end-to-end evaluation on real datasets, to better justify the need for dynamic adjustments.**
>
> The gLLM baseline adopts an even layer partitioning strategy.  We have also evaluated other static uneven partitioning strategies under real workloads, and Dynapipe consistently outperforms them.  This is because static strategies cannot adapt to the optimal layer distribution during runtime.
>
> Considering the wide variety of static uneven partitioning schemes, as well as our comparisons against other systems such as vLLM and SGLang, including all baselines in a single figure would lead to visual clutter and hinder readability.  Therefore, we designed an ablation study in Figure 5, where we vary the input/output length ratio to simulate different prefill/decode workload distributions.  This setup indirectly reflects how different static partitioning strategies perform under various load conditions.  The results further demonstrate that Dynapipe consistently outperforms these methods, highlighting the necessity and advantage of dynamic layer adjustment.
>
> This is particularly important in real-world serving environments, where the engine typically handles mixed workloads rather than a single dataset.  Requests from different applications arrive at different times with diverse input/output length patterns.  In such dynamic scenarios, Dynapipe can more effectively adapt the system to the current request distribution, delivering better and more stable performance.

---

> > ### Author Response · Authors · 2025-08-06
> >
> > Dear Reviewer,
> >
> > I hope this message finds you well.   As the discussion period is nearing its end with less than two days remaining, I wanted to ensure we have addressed all your concerns satisfactorily.   If there are any additional points or feedback you'd like us to consider, please let us know.   Your insights are invaluable to us, and we’re eager to address any remaining issues to improve our work.
> >
> > Thank you for your time and effort in reviewing our paper.

---

> > ### Comment · Reviewer_Zxmv · 2025-08-07
> >
> > I am satisfied with authors' response to most of my questions.
> >
> > For Q5 on "the system pre-allocates memory and loads the weights for additional layers that may be reassigned", a systematic approach on deciding how many/which layers to preload is needed.

---

> > > ### Author Response · Authors · 2025-08-07
> > >
> > > We sincerely thank the reviewer for raising the important issue regarding the systematic design of the preloading strategy. In our current implementation, we adopt a deterministic rule based on the pipeline parallel topology to decide which layers’ weights each GPU should preload.
> > >
> > > Specifically, the number of preloaded layers depends on the maximum potential layer redistribution range that the system might perform in the future. Take, for example, a Transformer model with 48 layers deployed across 4 GPUs using pipeline parallelism. In a standard equal partitioning, the layer distribution is as follows: GPU 0 handles layers 0–11, GPU 1 handles layers 12–23, GPU 2 handles layers 24–35, and GPU 3 handles layers 36–47. If we consider a maximum dynamic reallocation margin of 3 layers, the most extreme redistribution scenario could be: GPU 0 handling layers 0–12, GPU 1 handling layers 13–25, GPU 2 handling layers 26–38, and GPU 3 handling layers 39–47. Therefore, to accommodate future adjustments, each GPU needs to preload additional adjacent layers during system initialization. For instance, GPU 2 must additionally load layers 36–38.
> > >
> > > As illustrated below:
> > >
> > > ```txt
> > > Original balanced assignment:
> > > [GPU 0]  0 ── 11
> > > [GPU 1]          12 ── 23
> > > [GPU 2]                   24 ── 35
> > > [GPU 3]                            36 ── 47
> > >
> > > Dynapipe's preload range (only a subset is actively used at runtime):
> > > [GPU 0]  0 ─── 12
> > > [GPU 1]        12 ───────── 25
> > > [GPU 2]                   24 ─────── 38
> > > [GPU 3]                            36 ── 47
> > > ```
> > >
> > > > It is important to note that the layers preloaded by different GPUs may partially overlap. This is a deliberate design to enable future dynamic layer migration by preloading layers that might be borrowed, thus improving the system's responsiveness.
> > >
> > > In terms of implementation, suppose the model has a total of `num_layers`, and the system uses `pp_size` pipeline stages (i.e., number of GPUs), each stage is assigned `num_layers_pp = num_layers // pp_size` layers on average. The current preloading strategy is implemented as follows:
> > >
> > > ```python
> > > def get_preload_layer_range(num_layers: int, pp_size: int, rank: int):
> > >     # Return the layer range [start, end) that the current GPU should preload
> > >     num_layers_pp = num_layers // pp_size
> > >     rank = get_pp_rank()  # The current GPU's position in the pipeline (starting from 0)
> > >     if rank != pp_size - 1:
> > >         # All but the last stage preload additional layers
> > >         return num_layers_pp * rank, num_layers_pp * (rank + 1) + (rank + 1) % pp_size
> > >     else:
> > >         # The last stage loads only its own assigned layers
> > >         return num_layers_pp * rank, num_layers
> > > ```
> > >
> > > In this strategy:
> > >
> > > - The final stage (`rank = pp_size - 1`) only loads its own assigned layers;
> > > - All other stages preload an additional `(rank + 1) % pp_size` layers, depending on their position in the pipeline, to accommodate potential layer migration.
> > >
> > > In future work, we plan to adopt a more dynamic optimization strategy: rather than preloading all redundant weights into GPU memory at initialization, we will prioritize loading them into CPU memory. Combined with runtime execution prediction mechanisms, layers that are about to be used can then be asynchronously transferred to the GPU, while currently unused layers can be offloaded, thereby avoiding unnecessary GPU memory consumption and improving memory efficiency.
> > >
> > > We greatly appreciate your feedback, which has helped us recognize areas where the paper needs to be more clearly and precisely written. We will revise the manuscript accordingly in the final version. Meanwhile, we would sincerely appreciate it if you would consider revising your score to a more positive rating, or whether our responses have helped improve your assessment of our work. If you have any further questions or suggestions, we would be more than happy to continue the discussion.

---

### Official Review · Reviewer_icHW · 2025-06-30

**Clarity:** 3
**Significance:** 3
**Originality:** 3
**Rating:** 4
**Confidence:** 4

**Summary:**

This paper identifies the inefficiency caused by costly sampling operation in LLM inference. Specifically, in pipeline parallel inference, the last sampling step takes more time on computation, thus creating idle bubble in the pipeline stages. The authors present empirical evidences to show the non-negligible impact of sampling operation, comparing with the Transformer layer and argued that traditional approach which splits the Transformer layers evenly is efficiently. They thus propose DynaPipe which relies on a latency predictor to adaptively adjust the layer distribution among devices to achieve a better balanced computation. Together with a KV cache migrator, DynaPipe can serve the requests with much lower latency.

**Questions:**

1. Figure 2(b)'s experiments looks confusing. For 2(b), what is the layer execution time here? I assume for a specific token size, the per-layer execution time should be similar. It also applies to the sampling time. Why the ratio would be different? Is the token size changing in this Figure? And is it a reasonable setting to compare the sampling time with only one layer computation? In the evaluation, Qwen 32 B has 64 layers, and each GPU would hosts 16 Transformer layers in a even setup. I think it is fairer to compare the sampling time with a real per-device execution time.
2. The siginificance of Figure 2(a) and Figure 2(b) are also unclear. In a computation-bound setup, it is common knowledge that the duration grows linearly with the size of the tokens. I don't see a connection with the paper's idea.
3. Pipeline parallelism can be used in multi-node setting due to its lighter communication burden compared with Tensor parallelism. Is there any results on a larger-scale setting, like 4 nodes?

**Ethical Concerns:**

["NO or VERY MINOR ethics concerns only"]

**Final Justification:**

My concerns are properly addressed. I willl maintain my score.

**Limitations:**

Yes.

**Paper Formatting Concerns:**

No formatting issues.

**Quality:**

3

**Strengths And Weaknesses:**

# Strengths
1. The authors identify an interesting problem in LLM serving. Most of the existing work focuses on accelerating the computation in Transformer layer. Sample indeed is a inevitable step in inference. This problem is more critical in generative LLM.
2. DynaPipe proposes a clear design to efficiently adjust the layer distribution across devices to achieve a more balanced states.
# Weakness
1. Some of the experiments results require further clarification. See questions below.
2. The evaluation could be improved with a larger scale setup.

---

> ### Author Rebuttal · Authors · 2025-07-31
>
> Thank you for your valuable comments. We appreciate your feedback and would like to address your concerns as follows.
>
> **Q1:**
>
> To begin, we would like to clarify some background details. Our experiments adopt the chunked prefill strategy, which allows prefill and decode requests to be mixed within a single batch and processed jointly during inference. In this setting, the sampling overhead primarily depends on the number of decode requests within the batch, while the forward execution time of each Transformer layer depends on the total number of tokens to be processed (including both prefill and decode tokens).
>
> We now explain the motivation behind Figure 2, which serves to justify the need for dynamic layer redistribution, as proposed in our system.
>
> **(1) Why does the ratio between sampling time and per-layer forward time vary?**
>
> Figure 2(b) aims to quantify the relative cost of the sampling operation. Specifically, under a realistic inference workload, this figure shows a distribution of the ratio between sampling latency and the forward pass time of a single Transformer layer. From empirical observation, this ratio averages around 3.09, but is not constant—it varies dynamically across different time intervals and workload conditions, ranging approximately from 1 to 5.
>
> The reasons for this variability include:
>
> - Sampling overhead increases with the number of decode requests, as illustrated in Figure 2(a). This overhead is determined solely by the number of decode tokens $n_d$ in a batch, since only decode requests require sampling operations.
> - The per-layer forward time grows with the total number of tokens, including both prefill and decode tokens, as shown in Figure 2(c). That is, the forward time depends on $n_p + n_d$, where $n_p$ is the number of prefill tokens and $n_d$ is the number of decode tokens.
> - The total token count $n_p + n_d$ per batch is not fixed in real-world serving scenarios. Since requests arrive dynamically, the ratio $n_d / (n_p + n_d)$ fluctuates over time. For example, at the beginning, when there are mostly prefill tokens and few decode tokens, sampling overhead is minimal. However, in later stages—if no new prefill requests arrive and the batch consists primarily of decode tokens—sampling overhead becomes dominant.
>
> Given these factors, the ratio between sampling time and per-layer forward time varies dynamically during inference. This non-stationarity in workload characteristics renders static layer allocation suboptimal for maintaining balanced pipeline stage runtimes, and therefore motivates DynaPipe’s dynamic layer redistribution mechanism.
>
> **(2) Definition of “layer execution time” in Figure 2(b), and rationale for using a per-layer unit instead of per-device execution time:**
>
> In Figure 2(b), the term “layer execution time” refers to the average forward time of a single Transformer layer under current batch conditions.
>
> Since all layers share the same architecture and process the same batch of tokens, the forward time per layer remains uniform under consistent conditions. Thus, we choose the per-layer execution time as a standardized unit for modeling and reasoning about the relative cost of sampling.
>
> Moreover, the per-layer unit facilitates scheduling decisions in our system: specifically, DynaPipe's bubble-aware scheduler estimates how many layers must be removed from the final pipeline stage to offset the sampling latency. For example, if the sampling delay is equivalent to four layers of forward computation, we can offload three layers from the final stage and redistribute them upstream, achieving better load balance.
>
> This modeling approach is not only intuitive and lightweight, but also simplifies the logic for runtime adaptation. We emphasize that quantifying one layer versus multiple layers is fundamentally equivalent, with the primary difference being the granularity of expression. Therefore, the design choice in Figure 2(b) is meant to support scheduling formulation, and does not constitute an unfair or misleading comparison.
>
> **Q2:The siginificance of Figure 2(a) and Figure 2(b) are also unclear. In a computation-bound setup, it is common knowledge that the duration grows linearly with the size of the tokens. I don't see a connection with the paper's idea.**
>
> The purpose of Figure 2(a) is to show how the sampling overhead varies with the number of sampled tokens, while Figure 2(c) illustrates the relationship between the single-layer forward computation time and the number of tokens to be processed.  These figures reflect the concrete trends in the overheads of both computation stages under different workload conditions, providing a data-driven foundation for building accurate performance prediction models.
>
> Since the overheads of both stages vary with workload, Figure 2(b) further presents the distribution density of their time ratio, showing that the relative cost between sampling and single-layer forward computation is not constant, but fluctuates within a dynamic range.  This variability directly motivates our design of a dynamic layer redistribution strategy.
>
> **Q3:Pipeline parallelism can be used in multi-node setting due to its lighter communication burden compared with Tensor parallelism. Is there any results on a larger-scale setting, like 4 nodes?**
>
> Pipeline parallelism generally offers better scalability in multi-node environments due to its lower communication overhead compared to tensor parallelism. In our experiments, we conducted evaluations on a single-node PCIe setup, where the communication cost of tensor parallelism is relatively low. Nevertheless, even under such conditions that favor tensor parallelism, our pipeline-based approach still demonstrates superior performance.
>
> Due to hardware constraints, we were unable to conduct experiments on real multi-node clusters (e.g., 4 nodes). However, we performed a simulated multi-node experiment on a single machine with four GPUs by disabling NCCL's P2P, shared memory, and IB communication to emulate the communication overhead of cross-node setups. Under these simulated conditions, our approach still achieved promising results. For instance, under simulated multi-node settings on the Qwen-32B model and ShareGPT dataset, our DynaPipe system consistently reduced end-to-end latency by 10%–20% compared to gLLM across request rates of 12, 18, and 24, despite an overall increase in latency due to the emulated communication overhead.
>
> It is worth noting that the original gLLM paper has already demonstrated its superiority over tensor parallelism in real multi-node environments. Given that our DynaPipe outperforms gLLM in simulated multi-node settings, we have strong reason to believe that it would deliver even better performance than tensor parallelism in real-world multi-node deployments.

---

> > ### Comment · Reviewer_icHW · 2025-08-05
> >
> > If I understand correctly, Figure 2(b) is a end-to-end performance measurement, right? Assume several batch of request, the number of prefill and decode are varying. Thus making the ratio varying. It would be better to clarify the experiment setup in the paper.
> > For using the per-layer duration as the basic unit, I agree it matches with the paper's idea. But it would be a bit weak to say it is one of the bottleneck of the inference pipeline.

---

> > > ### Author Response · Authors · 2025-08-06
> > >
> > > Thank you very much for your response and suggestions. We would like to take this opportunity to clarify your concerns:
> > >
> > > - **Regarding Figure 2(b)**: This figure does not represent an end-to-end performance measurement. Instead, we conducted experiments using the Qwen2.5-32B model on four A100-40GB GPUs (PCIe) under a request rate of 18, evaluated on the ShareGPT dataset. The measured quantity is the overhead within the pipeline system **per microbatch** during the forward computation process—specifically, the **average time for a single Transformer layer's forward computation** (as previously explained, we use the time of a single-layer forward pass for estimation) and **the sampling overhead in the final stage**. Then, we compute the ratio of the sampling overhead to the single-layer forward time for each microbatch and visualize the **distribution of this ratio** in Figure 2(b).
> > >
> > >   As you correctly pointed out, the number of prefill and decode tokens varies dynamically, which leads to fluctuations in the sampling-to-forward time ratio. The **motivation** of this figure is to demonstrate that this ratio changes significantly during execution, indicating that a **static layer redistribution strategy** cannot adapt well to such dynamic workload. This, in turn, motivates our **dynamic layer redistribution approach**, which can adjust the number of layers assigned to the last GPU based on the real-time workload.
> > >
> > > - **On the bottleneck in inference pipelines**: The core bottleneck arises from the **imbalance of computation load across GPUs**—the last GPU is responsible for both forward computation and sampling, while the preceding GPUs only handle forward passes. This imbalance leads to **pipeline bubbles**, which accumulate and have an increasingly negative impact on subsequent requests. This performance degradation is the primary motivation for redistributing model layers. However, as discussed earlier, **static redistribution fails to adapt to runtime dynamics**, which is why we propose a **dynamic layer redistribution** strategy.
> > >
> > > We greatly appreciate your feedback, which has helped us recognize areas where the paper needs to be more clearly and precisely written. We will revise the manuscript accordingly in the final version. Meanwhile, we would like to kindly ask whether you would consider adjusting your score, or whether our responses have helped improve your assessment of our work. If you have any further questions or suggestions, we would be more than happy to continue the discussion.

---

### Note · Authors · 2025-08-13

We sincerely thank all reviewers for their valuable feedback and for highlighting the strengths of our work.

Reviewers generally acknowledged our key contributions:

- **Problem identification**: We reveal the inter-stage imbalance issue in pipeline systems caused by sampling latency.
- **Effective solution**: DynaPipe is clearly designed, easy to implement, and dynamically redistributes layers at runtime to mitigate pipeline bubbles.
- **Comprehensive evaluation**: Experimental results strongly validate the effectiveness of our approach.

------

The main concerns raised are as follows:

- **Motivation for dynamic layer redistribution**: As shown in Fig. 2, in LLM serving, sampling cost depends only on the number of decode requests. The varying ratio of prefill to decode requests over time changes the balance between forward and sampling costs, thus affecting the optimal number of layers to be migrated from the last GPU. Please refer to our reply to Reviewer Zxmv for detailed analysis.
- **Additional overhead**: Our method introduces communication cost and some memory overhead, which is a trade-off for significant performance gains, as shown in experiments. Communication cost is mitigated through computation–communication overlap and by filtering low-benefit adjustments. Memory overhead (~10–15%) is mainly from extra layer weights, which could be reduced in future work via asynchronous loading from CPU memory.
- **Distinction from training-focused work**: Some training works address imbalance but do not consider dynamic request patterns, and often require strict communication synchronization, making them unsuitable for inference scenarios. See our reply to Reviewer oYmA for details.

------

In the revised version, we plan to:

1. Add experiments on more models and  quantify the overhead of each component.
2. Strengthen the Related Work section by comparing with training-focused studies, and include overhead discussion in the Limitation section.
3. Discuss in Future Work how to reduce current overhead using offloading techniques.

---

### Decision · Program_Chairs · 2025-09-17

**Decision:**

Accept (poster)

**Comment:**

Summary: The paper identifies a practically important bottleneck in pipeline-parallel LLM serving and presents a simple, implementable mechanism (dynamic layer redistribution with async KV migration and bubble-aware scheduling) that delivers consistent latency reductions, with reasonable overheads and clear engineering.

Strengths: 1. Sampling-time bubbles at the tail stage are real and under-addressed in serving stacks. Reviewers agree the observation is interesting/important.
2. Strong speedups (8–49%) with explanations for when/why gains peak; code provided and experiments reproducible

Weakness: 1. Novelty is incremental relative to training literature on uneven partitions and communication overlap; the contribution is primarily in porting/adapting these ideas to inference with runtime dynamics.
2. Only two Qwen variants; reviewers requested more sizes and possibly MoE/dense diversity (authors agree to expand)

Decision: Works with common serving patterns (chunked prefill) and shows benefits even in simulated multi-node settings over a strong pipeline baseline (gLLM).  The paper surfaces a real bottleneck and shows a clean, deployable fix with consistent gains across rates/lengths.

Discussion Summary: Authors clarified per-layer forward time as a normalized unit for modeling/scheduling; the scheduler decides how many layers to shift based on this unit. They argued training not same as inference (backward phase, synchronization) and emphasized runtime dynamics; they promised more overhead breakdowns. Main concerns addressed. Not all points raised by oYma addressed on Dynapipe vs Vocabulary parallelism, but can be addressed through future works. Cross review concern on overheads: Authors quantified memory overhead (~10–15%; ~16% in Qwen-32B example) and explained how async overlap + window-threshold=25 restrain migration costs.